# SPHERICAL CNNS ON UNSTRUCTURED GRIDS

**Chiyu "Max" Jiang**
UC Berkeley

**Jingwei Huang**
Stanford University

**Karthik Kashinath**
Lawrence Berkeley Nat'l Lab

**Prabhat**
Lawrence Berkeley Nat'l Lab

**Philip Marcus**
UC Berkeley

**Matthias Nießner**
Technical University of Munich

## ABSTRACT

We present an efficient convolution kernel for Convolutional Neural Networks (CNNs) on unstructured grids using parameterized differential operators while focusing on spherical signals such as panorama images or planetary signals. To this end, we replace conventional convolution kernels with linear combinations of differential operators that are weighted by learnable parameters. Differential operators can be efficiently estimated on unstructured grids using one-ring neighbors, and learnable parameters can be optimized through standard back-propagation. As a result, we obtain extremely efficient neural networks that match or outperform state-of-the-art network architectures in terms of performance but with a significantly smaller number of network parameters. We evaluate our algorithm in an extensive series of experiments on a variety of computer vision and climate science tasks, including shape classification, climate pattern segmentation, and omnidirectional image semantic segmentation. Overall, we (1) present a novel CNN approach on unstructured grids using parameterized differential operators for spherical signals, and (2) show that our unique kernel parameterization allows our model to achieve the same or higher accuracy with significantly fewer network parameters.

## 1 INTRODUCTION

A wide range of machine learning problems in computer vision and related areas require processing signals in the spherical domain; for instance, omnidirectional RGBD images from commercially available panorama cameras, such as Matterport (Chang et al., 2017), panaramic videos coupled with LIDAR scans from self-driving cars (Geiger et al., 2013), or planetary signals in scientific domains such as climate science (Racah et al., 2017). Unfortunately, naively mapping spherical signals to planar domains results in undesirable distortions. Specifically, projection artifacts near polar regions and handling of boundaries makes learning with 2D convolutional neural networks (CNNs) particularly challenging and inefficient. Very recent work, such as Cohen et al. (2018) and Esteves et al. (2018), propose network architectures that operate natively in the spherical domain, and are invariant to rotations in the $\mathcal{SO}(3)$ group. Such invariances are desirable in a set of problems – e.g., machine learning problems of molecules – where gravitational effects are negligible and orientation is arbitrary. However, for other different classes of problems at large, assumed orientation information is crucial to the predictive capability of the network. A good example of such problems is the MNIST digit recognition problem, where orientation plays an important role in distinguishing digits "6" and "9". Other examples include omnidirectional images, where images are naturally oriented by gravity; and planetary signals, where planets are naturally oriented by their axis of rotation.

In this work, we present a new convolution kernel for CNNs on arbitrary manifolds and topologies, discretized by an unstructured grid (i.e., mesh), and focus on its applications in the spherical domain approximated by an icosahedral spherical mesh. We propose and evaluate the use of a new parameterization scheme for CNN convolution kernels, which we call Parameterized Differential Operators (PDOs), which is easy to implement on unstructured grids. We call the resulting convolution operator that operates on the mesh using such kernels the MeshConv operator. This parameterization scheme utilizes only 4 parameters for each kernel, and achieves significantly better performance

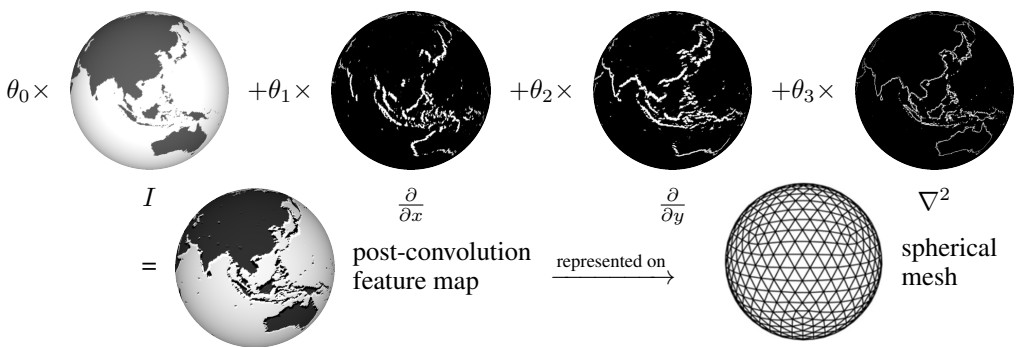

Figure 1: Illustration for the MeshConv operator using parameterized differential operators to replace conventional learnable convolutional kernels. Similar to classic convolution kernels that establish patterns between neighboring values, differential operators computes "differences", and a linear combination of differential operators establishes similar patterns.

than competing methods, with much fewer parameters. In particular, we illustrate its use in various machine learning problems in computer vision and climate science.

In summary, our contributions are as follows:

- We present a general approach for orientable CNNs on unstructured grids using parameterized differential operators.
- We show that our spherical model achieves significantly higher parameter efficiency compared to state-of-the-art network architectures for 3D classification tasks and spherical image semantic segmentation.
- We release and open-source the codes developed and used in this study for other potential extended applications[1].

We organize the structure of the paper as follows. We first provide an overview of related studies in the literature in Sec. 2; we then introduce details of our methodology in Sec. 3, followed by an empirical assessment of the effectiveness of our model in Sec. 4. Finally, we evaluate the design choices of our kernel parameterization scheme in Sec. 5.

## 2 BACKGROUND

**Spherical CNNs** The first and foremost concern for processing spherical signals is distortions introduced by projecting signals on curved surfaces to flat surfaces. Su & Grauman (2017) process equirectangular images with regular convolutions with increased kernel sizes near polar regions where greater distortions are introduced by the planar mapping. Coors et al. (2018) and Zhao et al. (2018) use a constant kernel that samples points on the tangent plane of the spherical image to reduce distortions. A slightly different line of literature explores rotational-equivariant implementations of spherical CNNs. Cohen et al. (2018) proposed spherical convolutions with intermediate feature maps in $\mathcal{SO}(3)$ that are rotational-equivariant. Esteves et al. (2018) used spherical harmonic basis to achieve similar results.

**Reparameterized Convolutional Kernel** Related to our approach in using parameterized differential operators, several works utilize the diffusion kernel for efficient Machine Learning and CNNs. Kondor & Lafferty (2002) was among the first to suggest the use of diffusion kernel on graphs. Atwood & Towsley (2016) propose Diffusion-Convolutional Neural Networks (DCNN) for efficient convolution on graph structured data. Boscaini et al. (2016) introduce a generalization of classic CNNs to non-Euclidean domains by using a set of oriented anisotropic diffusion kernels. Cohen & Welling (2016) utilized a linear combination of filter banks to acquire equivariant convolution filters.

---

[1] Our codes are available on Github: `https://github.com/maxjiang93/ugscnn`

Ruthotto & Haber (2018) explore the reparameterization of convolutional kernels using parabolic and hyperbolic differential basis with regular grid images.

**Non-Euclidean Convolutions**   Related to our work on performing convolutions on manifolds represented by an unstructured grid (i.e., mesh), works in geometric deep learning address similar problems (Bronstein et al., 2017). Other methods perform graph convolution by parameterizing the convolution kernels in the spectral domain, thus converting the convolution step into a spectral dot product (Bruna et al., 2014; Defferrard et al., 2016; Kipf & Welling, 2017; Yi et al., 2017). Masci et al. (2015) perform convolutions directly on manifolds using cross-correlation based on geodesic distances and Maron et al. (2017) use an optimal surface parameterization method (seamless toric covers) to parameterize genus-zero shapes into 2D signals for analysis using conventional planar CNNs.

**Image Semantic Segmentation**   Image semantic segmentation is a classic problem in computer vision, and there has been an impressive body of literature studying semantic segmentation of planar images (Ronneberger et al., 2015; Badrinarayanan et al., 2015; Long et al., 2015; Jégou et al., 2017; Wang et al., 2018a). Song et al. (2017) study semantic segmentation of equirectangular omnidirectional images, but in the context of image inpainting, where only a partial view is given as input. Armeni et al. (2017) and Chang et al. (2017) provide benchmarks for semantic segmentation of 360 panorama images. In the 3D learning literature, researchers have looked at 3D semantic segmentation on point clouds or voxels (Dai et al., 2017a; Qi et al., 2017a; Wang et al., 2018b; Tchapmi et al., 2017; Dai et al., 2017b). Our method also targets the application domain of image segmentation by providing a more efficient convolutional operator for spherical domains, for instance, focusing on panoramic images (Chang et al., 2017).

## 3   METHOD

### 3.1   PARAMETERIZED DIFFERENTIAL OPERATORS

We present a novel scheme for efficiently performing convolutions on manifolds approximated by a given underlying mesh, using what we call Parameterized Differential Operators. To this end, we reparameterize the learnable convolution kernel as a linear combination of differential operators. Such reparameterization provides two distinct advantages: first, we can drastically reduce the number of parameters per given convolution kernel, allowing for an efficient and lean learning space; second, as opposed to the cross-correlation type convolution on mesh surfaces (Masci et al., 2015), which requires large amounts of geodesic computations and interpolations, first and second order differential operators can be efficiently estimated using only the one-ring neighborhood.

In order to illustrate the concept of PDOs, we draw comparisons to the conventional $3 \times 3$ convolution kernel in the regular grid domain. The $3 \times 3$ kernel parameterized by parameters $\boldsymbol{\theta}$: $\mathcal{G}_{\boldsymbol{\theta}}^{3 \times 3}$ can be written as a linear combination of basis kernels which can be viewed as delta functions at constant offsets:

$$\mathcal{G}_{\boldsymbol{\theta}}^{3 \times 3}(x, y) = \sum_{i=-1}^{1} \sum_{j=-1}^{1} \theta_{ij} \delta(x - i, y - j) \tag{1}$$

where $x$ and $y$ refer to the spatial coordinates that correspond to the two spatial dimensions over which the convolution is performed. Due to the linearity of the cross-correlation operator ($*$), the output feature map can be expressed as a linear combination of the input function cross-correlated with different basis functions. Defining the linear operator $\Delta_{ij}$ to be the cross-correlation with a basis delta function, we have:

$$\Delta_{ij} \mathcal{F}(x, y) := \mathcal{F}(x, y) * \delta(x - i, y - j) \tag{2}$$

$$\mathcal{F}(x, y) * \mathcal{G}_{\boldsymbol{\theta}}^{3 \times 3}(x, y) = \sum_{i=-1}^{1} \sum_{j=-1}^{1} \theta_{ij} \Delta_{ij} \mathcal{F}(x, y) \tag{3}$$

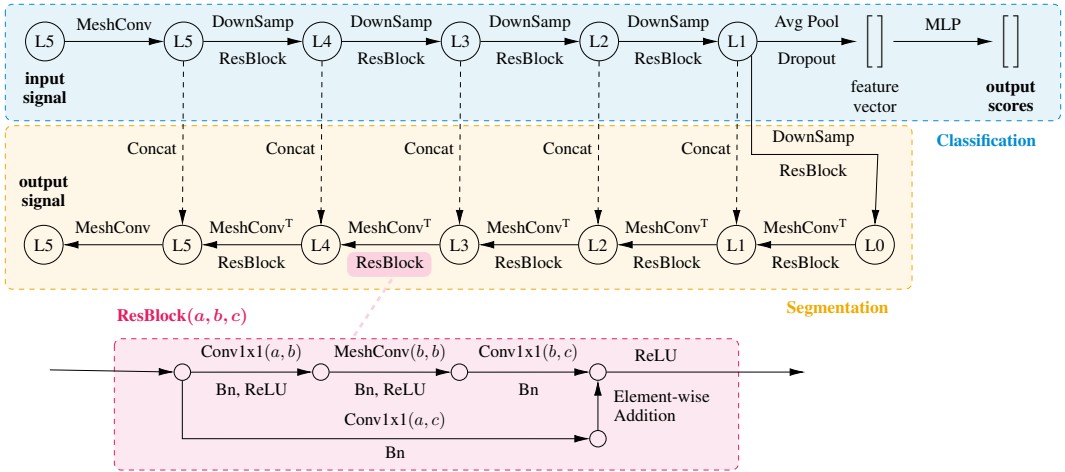

Figure 2: Schematics for model architecture for classification and semantic segmentation tasks, at a level-5 input resolution. L$n$ stands for spherical mesh of level-$n$ as defined in Sec. 3.2. MeshConv is implemented according to Eqn. 4. MeshConv$^T$ first pads unknown values at the next level with 0, followed by a regular MeshConv. DownSamp samples the values at the nodes in the next mesh level. A ResBlock with bottleneck layers, consisting of Conv1x1 (1-by-1 convolutions) and MeshConv layers is detailed above. In the decoder, ResBlock is after each MeshConv$^T$ and Concat.

In our formulation of PDOs, we replace the cross-correlation linear operators $\Delta_{ij}$ with differential operators of varying orders. Similar to the linear operators resulting from cross-correlation with basis functions, differential operators are linear, and approximate local features. In contrast to cross-correlations on manifolds, differential operators on meshes can be efficiently computed using Finite Element basis, or derived by Discrete Exterior Calculus. In the actual implementation below, we choose the identity ($I$, 0th order differential, same as $\Delta_{00}$), derivatives in two orthogonal spatial dimensions ($\nabla_x, \nabla_y$, 1st order differential), and the Laplacian operator ($\nabla^2$, 2nd order differential):

$$\mathcal{F}(x,y) * \mathcal{G}_{\boldsymbol{\theta}}^{diff} = \theta_0 I\mathcal{F} + \theta_1 \nabla_x \mathcal{F} + \theta_2 \nabla_y \mathcal{F} + \theta_3 \nabla^2 \mathcal{F} \tag{4}$$

The identity ($I$) of the input function is trivial to obtain. The first derivative ($\nabla_x, \nabla_y$) can be obtained by first computing the per-face gradients, and then using area-weighted average to obtain per-vertex gradient. The dot product between the per-vertex gradient value and the corresponding $x$ and $y$ vector fields are then computed to acquire $\nabla_x \mathcal{F}$ and $\nabla_y \mathcal{F}$. For the sphere, we choose the east-west and north-south directions to be the $x$ and $y$ components, since the poles naturally orient the spherical signal. The Laplacian operator on the mesh can be discretized using the cotangent formula:

$$\nabla^2 \mathcal{F} \approx \frac{1}{2\mathcal{A}_i} \sum_{j \in \mathcal{N}(i)} (\cot \alpha_{ij} + \cot \beta_{ij})(\mathcal{F}_i - \mathcal{F}_j) \tag{5}$$

where $\mathcal{N}(i)$ is the nodes in the neighboring one-ring of $i$, $\mathcal{A}_i$ is the area of the dual face corresponding to node $i$, and $\alpha_{ij}$ and $\beta_{ij}$ are the two angles opposing edge $ij$. With this parameterization of the convolution kernel, the parameters can be similarly optimized via backpropagation using standard stochastic optimization routines.

## 3.2 Icosahedral Spherical Mesh

The icosahedral spherical mesh (Baumgardner & Frederickson, 1985) is among the most uniform and accurate discretizations of the sphere. A spherical mesh can be obtained by progressively subdividing each face of the unit icosahedron into four equal triangles and reprojecting each node to unit distance from the origin. Apart from the uniformity and accuracy of the icosahedral sphere, the subdivision scheme for the triangles provides a natural coarsening and refinement scheme for the

| Model | Accuracy(%) | Number of Parameters |
|-------|-------------|----------------------|
| S2CNN (Cohen et al., 2018) | 96.00 | 58k |
| SphereNet (Coors et al., 2018) | 94.41 | 196k |
| Ours | **99.23** | 62k |

Table 1: Results on the Spherical MNIST dataset for validating the use of Parameterized Differential Operators. Our model achieves state-of-the-art performance with comparable number of training parameters.

grid that allows for easy implementations of pooling and unpooling routines associated with CNN architectures. See Fig. 1 for a schematic of the level-3 icosahedral spherical mesh.

For the ease of discussion, we adopt the following naming convention for mesh resolution: starting with the unit icosahedron as the level-0 mesh, each progressive mesh resolution is one level above the previous. Hence, for a level-$l$ mesh:

$$n_f = 20 \cdot 4^l; n_e = 30 \cdot 4^l; n_v = n_e - n_f + 2 \tag{6}$$

where $n_f, n_e, n_v$ stands for the number of faces, edges, and vertices of the spherical mesh.

### 3.3 MODEL ARCHITECTURE DESIGN

A detailed schematic for the neural architectures in this study is presented in Fig. 2. The schematic includes architectures for both the classification and regression network, which share a common encoder architecture. The segmentation network consists of an additional decoder which features transpose convolutions and skip layers, inspired by the U-Net architecture (Ronneberger et al., 2015). Minor adjustments are made for different tasks, mainly surrounding adjusting the number of input and output layers to process signals at varied resolutions. A detailed breakdown for model architectures, as well as training details for each task in the Experiment section (Sec. 4), is provided in the appendix (Appendix Sec. B).

## 4 EXPERIMENTS

### 4.1 SPHERICAL MNIST

To validate the use of parameterized differential operators to replace conventional convolution operators, we implemented such neural networks towards solving the classic computer vision benchmark task: the MNIST digit recognition problem (LeCun, 1998).

**Experiment Setup**   We follow Cohen et al. (2018) by projecting the pixelated digits onto the surface of the unit sphere. We further move the digits to the equator to prevent coordinate singularity at the poles. We benchmark our model against two other implementations of spherical CNNs: a rotational-invariant model by Cohen et al. (2018) and an orientable model by Coors et al. (2018). All models are trained and tested with non-rotated digits to illustrate the performance gain from orientation information.

**Results and Discussion**   Our model outperforms its counterparts by a significant margin, achieving the best performance among comparable algorithms, with comparable number of parameters. We attribute the success in our model to the gain in orientation information, which is indispensable for many vision tasks. In contrast, S2CNN (Cohen et al., 2018) is rotational-invariant, and thus has difficulties distinguishing digits "6" and "9".

### 4.2 3D OBJECT CLASSIFICATION

We use the ModelNet40 benchmark (Wu et al., 2015), a 40-class 3D classification problem, to illustrate the applicability of our spherical method to a wider set of problems in 3D learning. For this study, we look into two aspects of our model: peak performance and parameter efficiency.

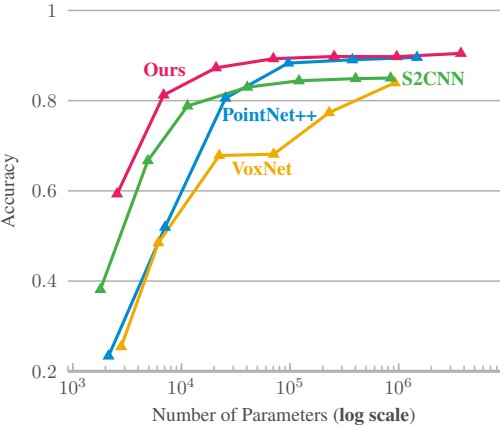 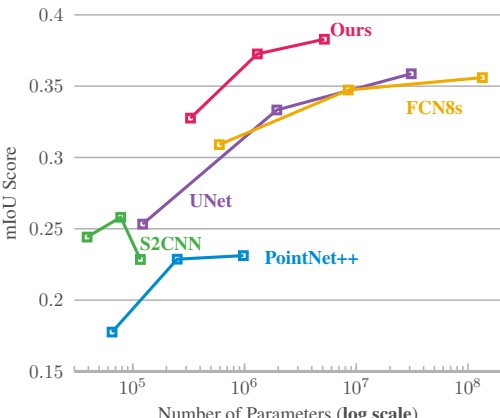

Figure 3: Parameter efficiency study on Model-Net40, benchmarked against representative 3D learning models consuming different input data representations: PointNet++ using point clouds as input, VoxNet consuming binary-voxel inputs, S2CNN consuming the same input structure as our model (spherical signal). The abscissa is drawn based on log scale.

Figure 4: Parameter efficiency study on 2D3DS semantic segmentation task. Our spherical segmentation model outperforms the planar and point-based counterparts by a significant margin across all parameter regimes.

**Experiment Setup** To use our spherical CNN model for the object classification task, we pre-process the 3D geometries into spherical signals. We follow Cohen et al. (2018) for preprocessing the 3D CAD models. First, we normalize and translate each mesh to the coordinate origin. We then encapsulate each mesh with a bounding level-5 unit sphere and perform ray-tracing from each point to the origin. We record the distance from the spherical surface to the mesh, as well as the $\sin, \cos$ of the incident angle. The data is further augmented with the 3 channels corresponding to the convex hull of the input mesh, forming a total of 6 input channels. An illustration of the data pre-processing process is presented in Fig. 5. For peak performance, we compare the best performance achievable by our model with other 3D learning algorithms. For the parameter efficiency study, we progressively reduce the number of feature layers in all models without changing the overall model architecture. Then, we evaluate the models after convergence in 250 epochs. We benchmark our results against PointNet++ (Qi et al., 2017a), VoxNet (Qi et al., 2016), and S2CNN[2].

**Results and Discussion** Fig. 3 shows a comparison of model performance versus number of parameters. Our model achieves the best performance across all parameter ranges. In the low-parameter range, our model is able to achieve approximately $60\%$ accuracy for the 40-class 3D classification task with a mere $2000+$ parameters. Table 2 shows a comparison of peak performance between models. At peak performance, our model is on-par with comparable state-of-the-art models, and achieves the best performance among models consuming spherical input signals.

### 4.3 Omnidirectional Image Segmentation

We illustrate the semantic segmentation capability of our network on the omnidirectional image segmentation task. We use the Stanford 2D3DS dataset (Armeni et al., 2017) for this task. The 2D3DS dataset consists of 1,413 equirectangular images with RGB+depth channels, as well as semantic labels across 13 different classes. The panoramic images are taken in 6 different areas, and the dataset is officially split for a 3-fold cross validation. While we are unable to find reported results on the semantic segmentation of these omnidirectional images, we benchmark our spherical segmentation algorithm against classic 2D image semantic segmentation networks as well as a 3D point-based model, trained and evaluated on the same data.

---

[2]We use the author's open-source implementations: PointNet++, VoxNet, S2CNN. We run PointNet++ with standard input of 1000 points with xyz coordinates for the classification task.

| Model | Input | Accu. (%) |
|---|---|---|
| 3DShapeNets (Wu et al., 2015) | voxels | 84.7 |
| VoxNet (Maturana & Scherer, 2015) | voxels | 85.9 |
| PointNet (Qi et al., 2017a) | points | 89.2 |
| PointNet++ (Qi et al., 2017b) | points | 91.9 |
| DGCNN (Wang et al., 2018b) | points | **92.2** |
| S2CNN (Cohen et al., 2018) | spherical | 85.0 |
| SphericalCNN (Esteves et al., 2018) | spherical | 88.9 |
| Ours | spherical | 90.5 |

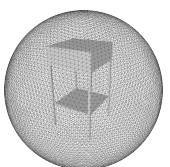
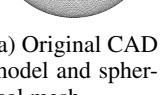
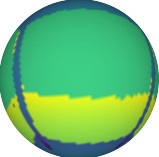

(a) Original CAD model and spherical mesh.

(b) Resulting surface distance signal.

Table 2: Results on ModelNet40 dataset. Our method compares favorably with state-of-the-art, and achieves best performance among networks utilizing spherical input signals.

Figure 5: Illustration of spherical signal rendering process for a given 3D CAD model.

**Experiment Setup** First, we preprocess the data into a spherical signal by sampling the original rectangular images at the latitude-longitudes of the spherical mesh vertex positions. Input RGB-D channels are interpolated using bilinear interpolation, while semantic labels are acquired using nearest-neighbor interpolation. We input and output spherical signals at the level-5 resolution. We use the official 3-fold cross validation to train and evaluate our results. We benchmark our semantic segmentation results against two classic semantic segmentation networks: the U-Net (Ronneberger et al., 2015) and FCN8s (Long et al., 2015). We also compared our results with a modified version of spherical S2CNN, and 3D point-based method, PointNet++ (Qi et al., 2017b) using $(x, y, z$,r,g,b) inputs reconstructed from panoramic RGBD images. We provide additional details toward the implementation of these models in Appendix E. We evaluate the network performance under two standard metrics: mean Intersection-over-Union (mIoU), and pixel-accuracy. Similar to Sec. 4.2, we evaluate the models under two settings: peak performance and a parameter efficiency study by varying model parameters. We progressively decimate the number of feature layers uniformly for all models to study the effect of model complexity on performance.

**Results and Discussion** Fig. 4 compares our model against state-of-the-art baselines. Our spherical segmentation outperforms the planar baselines for all parameter ranges, and more significantly so compared to the 3D PointNet++. We attribute PointNet++'s performance to the small amount of training data. Fig. 6 shows a visualization of our semantic segmentation performance compared to the ground truth and the planar baselines.

### 4.4 CLIMATE PATTERN SEGMENTATION

To further illustrate the capabilities of our model, we evaluate our model on the climate pattern segmentation task. We follow Mudigonda et al. (2017) for preprocessing the data and acquiring the ground-truth labels for this task. This task involves the segmentation of Atmospheric Rivers (AR) and Tropical Cyclones (TC) in global climate model simulations. Following Mudigonda et al. (2017), we analyze outputs from a 20-year run of the *Community Atmospheric Model v5 (CAM5)* (Neale et al., 2010). We benchmark our performance against Mudigonda et al. (2017) for the climate segmentation task to highlight our model performance. We preprocess the data to level-5 resolution.

| Model | Background (%) | TC (%) | AR (%) | Mean (%) |
|---|---|---|---|---|
| Mudigonda et al. (2017) | 97 | 74 | 65 | 78.67 |
| Ours | 97 | **94** | **93** | **94.67** |

Table 3: We achieves better accuracy compared to our baseline for climate pattern segmentation.

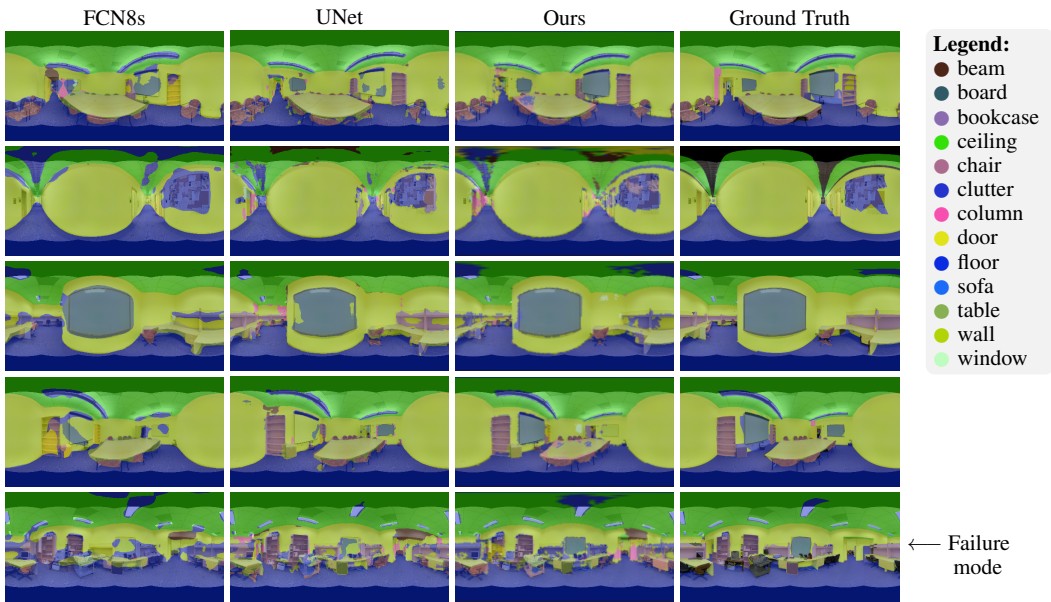

Figure 6: Visualization of semantic segmentation results on test set. Our results are generated on a level-5 spherical mesh and mapped to the equirectangular grid for visualization. Model underperforms in complex environments, and fails to predict ceiling lights due to incomplete RGB inputs.

**Results and Discussion**    Segmentation accuracy is presented in Table 3. Our model achieves better segmentation accuracy as compared to the baseline models. The baseline model (Mudigonda et al., 2017) trains and tests on random crops of the global data, whereas our model inputs the entire global data and predicts at the same output resolution as the input. Processing full global data allows the network to acquire better holistic understanding of the information, resulting in better overall performance.

## 5    ABLATION STUDY

We further perform an ablation study for justifying the choice of differential operators for our convolution kernel (as in Eqn. 4). We use the ModelNet40 classification problem as a toy example and use a 250k parameter model for evaluation. We choose various combinations of differential operators, and record the final classification accuracy. Results for the ablation study is presented in Table 4. Our choice of differential operator combinations in Eqn. 4 achieves the best performance

| Convolution kernel | Accuracy |
|---|---|
| $I + \frac{\partial}{\partial y} + \nabla^2$ | 0.8748 |
| $I + \frac{\partial}{\partial x} + \nabla^2$ | 0.8809 |
| $I + \nabla^2$ | 0.8801 |
| $I + \frac{\partial}{\partial x} + \frac{\partial}{\partial y}$ | 0.8894 |
| $I + \frac{\partial}{\partial x} + \frac{\partial}{\partial y} + \nabla^2$ | **0.8979** |

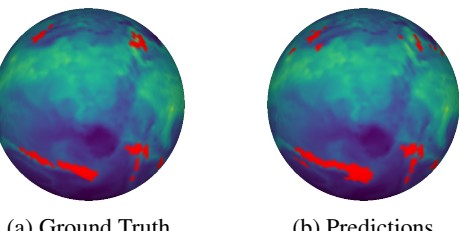

(a) Ground Truth        (b) Predictions

Table 4: Results for the ablation study. The choice of kernel that includes all differential operator components achieve the best accuracy, validating our choice of kernel in Eqn. 4.

Figure 7: Visualization of segmentation for Atmospheric River (AR). Plotted in the background is Integrated Vapor Transport (IVT), whereas red masks indicates the existance of AR.

among other choices, and the network performance improves with increased differential operators, thus allowing for more degrees of freedom for the kernel.

## 6 CONCLUSION

We have presented a novel method for performing convolution on unstructured grids using parameterized differential operators as convolution kernels. Our results demonstrate its applicability to machine learning problems with spherical signals and show significant improvements in terms of overall performance and parameter efficiency. We believe that these advances are particularly valuable with the increasing relevance of omnidirectional signals, for instance, as captured by real-world 3D or LIDAR panorama sensors.

## ACKNOWLEDGEMENTS

We would like to thank Taco Cohen for helping with the S2CNN comparison, Mayur Mudigonda, Ankur Mahesh, and Travis O'Brien for helping with the climate data, and Luna Huang for LATEXmagic. Chiyu "Max" Jiang is supported by the National Energy Research Scientific Computer (NERSC) Center summer internship program at Lawrence Berkeley National Laboratory. Prabhat and Karthik Kashinath are partly supported by the Intel Big Data Center. The authors used resources of NERSC, a DOE Office of Science User Facility supported by the Office of Science of the U.S. Department of Energy under Contract No. DE-AC02-05CH11231. In addition, this work is supported by a TUM-IAS Rudolf Mößbauer Fellowship and the ERC Starting Grant *Scan2CAD* (804724).

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

# APPENDIX

## A   ADDITIONAL DETAILS FOR IMPLEMENTING MESHCONV OPERATOR

In this section we provide more mathematical details for the implementation of the MeshConv Operator as described in Sec 3.1. In particular, we will describe in details the implementation of the various differential operators in Eqn. 4.

**Identity Operator**   The identity operator as suggested by its definition is just the identical replica of the original signal, hence no additional computation is required other than using the original signal as is.

**Gradient Operator**   Using a triangular mesh for discretizing the surface manifold, scalar functions on the surface can be discritized as a piecewise linear function, where values are defined on each vertex. Denoting the spatial coodinate vector as $\boldsymbol{x}$, the scalar function as $f(\boldsymbol{x})$, the scalar function values stored on vertex $i$ as $f_i$, and the piecewise linear "hat" functions as $\phi_i(\boldsymbol{x})$, we have:

$$f(\boldsymbol{x}) = \sum_{i=1}^{n} \phi_i(\boldsymbol{x}) f_i \tag{7}$$

the piecewise linear basis function $\phi_i(\boldsymbol{x})$ takes the value of $1$ on vertex $i$ and takes the value of $0$ on all other vertices. Hence, the gradient of this piecewise linear function can be computed as:

$$\nabla f(\boldsymbol{x}) = \nabla \sum_{i}^{n} \phi_i(\boldsymbol{x}) f_i = \sum_{i=1}^{n} \nabla \phi_i(\boldsymbol{x}) f_i \tag{8}$$

Due to the linearity of the basis functions $\phi_i$, the gradient is constant within each individual triangle. The per-face gradient value can be computed with a single linear operator $\boldsymbol{G}$ on the per-vertex scalar function $\boldsymbol{f}$:

$$\nabla f^{(f)} = \boldsymbol{G} \boldsymbol{f}^{(v)} \tag{9}$$

where the resulting per-face gradient is a 3-dimensional vector. We use the superscript $(f), (v)$ to distinguish per-face and per-vertex quantities. We refer the reader to Botsch et al. (2010) for detailed derivations for the gradient operator $G$. Denoting the area of each face as $\boldsymbol{a}^{(f)}$ (which can be easily computed using Heron's formula given coordinates of three points), the resulting per-vertex gradient vector can be computed as an average of per-face gradients, weighted by face area:

$$\nabla f_i^{(v)} = \frac{\sum_{j \in \mathcal{N}(i)} a_j^{(f)} \nabla f_j^{(f)}}{\sum_{j \in \mathcal{N}(i)} a_j^{(f)}} \tag{10}$$

denote the per-vertex polar (east-west) and azimuthal (north-south) direction fields as $\hat{\boldsymbol{x}}^{(v)}$ and $\hat{\boldsymbol{y}}^{(v)}$. They can be easily computed using the gradient operators detailed above, with the longitudinal and latitudinal values as the scalar function, followed by normalizing each vector to unit length. Hence two per-vertex gradient components can be computed as a dot-product against the unit directional fields:

$$\nabla_x f^{(v)} = \nabla f^{(v)} \cdot \hat{\boldsymbol{x}}^{(v)} \tag{11}$$

$$\nabla_y f^{(v)} = \nabla f^{(v)} \cdot \hat{\boldsymbol{y}}^{(v)} \tag{12}$$

**Laplacian Operator**   The mesh Laplacian operator is a standard operator in computational and differential geometry. We consider its derivation beyond the scope of this study. We provide the cotangent formula for computing the mesh Laplacian in Eqn. 5. We refer the reader to Chapters 6.2 and 6.3 of Crane (2015) for details of the derivation.

| Notation | Meaning |
|---|---|
| MeshConv($a, b$) | Mesh convolution layer with $a$ input channels and producing $b$ output channels |
| MeshConv($a, b$)$^{\text{T}}$ | Mesh transpose convolution layer with $a$ input channels and producing $b$ output channels. |
| BN | Batch Normalization. |
| ReLU | Rectified Linear Unit activation function |
| DownSamp | Downsampling spherical signal at the next resolution level. |
| ResBlock($a, b, c$) | As illustrated in Fig. 1, where $a, b, c$ stands for input channels, bottle neck channels, and output channels. |
| [ ]$_{\text{L}i}$ | The layers therein is at a mesh resolution of L$i$. |
| Concat | Concatenate skip layers of same resolution. |

Table 5: Network architecture notation list

## B  NETWORK ARCHITECTURE AND TRAINING DETAILS

In this section we provide detailed network architecture and training parameters for reproducing our results in Sec. 4. We use Fig. 2 as a reference.

### B.1  SPHERICAL MNIST

**Architecture**    Since the input signal for this experiment is on a level-4 mesh, the input pathway is slightly altered. The network architecture is as follows:

[MeshConv(1,16) + BN + ReLU]$_{\text{L4}}$ + [DownSamp + ResBlock(16, 16, 64)]$_{\text{L3}}$ + [DownSamp + ResBlock(64, 64, 256)]$_{\text{L2}}$ + AvgPool + MLP(256, 10)

Total number of parameters: 61658

**Training details**    We train our network with a batch size of 16, initial learning rate of $1 \times 10^{-2}$, step decay of 0.5 per 10 epochs, and use the Adam optimizer. We use the cross-entropy loss for training the classification network.

### B.2  3D OBJECT CLASSIFICATION

**Architecture**    The input signal is at a level-5 resolution. The network architecture closely follows that in the schematics in Fig. 2. We present two network architectures, one that corresponds to the network architecture with the highest accuracy score (the full model), and another that scales well with low parameter counts (the lean model). The full model:

[MeshConv(6, 32) + BN + ReLU]$_{\text{L5}}$ + [DownSamp + ResBlock(32, 32, 128)]$_{\text{L4}}$ + [DownSamp + ResBlock(128, 128, 512)]$_{\text{L3}}$ + [DownSamp + ResBlock(512, 512, 2048)]$_{\text{L2}}$ + AvgPool + MLP(2048, 40)

Total number of parameters: 3737160

The lean model:

[MeshConv(6, 8) + BN + ReLU]$_{\text{L5}}$ + [DownSamp + ResBlock(8, 8, 16)]$_{\text{L4}}$ + [DownSamp + ResBlock(16, 16, 64)]$_{\text{L3}}$ + [DownSamp + ResBlock(64, 64, 256)]$_{\text{L2}}$ + AvgPool + MLP(256, 40)

Total number of parameters: 70192

**Training details**    We train our network with a batch size of 16, initial learning rate of $5 \times 10^{-3}$, step decay of 0.7 per 25 epochs, and use the Adam optimizer. We use the cross-entropy loss for training the classification network.

## B.3 OMNIDIRECTIONAL IMAGE SEGMENTATION

**Architecture** Input signal is sampled at a level-5 resolution. The network architecture is identical to the segmentation network in Fig. 2. Encoder parameters are as follows:

[MeshConv(4,32)]$_{L5}$ + [DownSamp + ResBlock(32, 32, 64)]$_{L4}$ + [DownSamp + ResBlock(64, 64, 128)]$_{L3}$ + [DownSamp + ResBlock(128, 128, 256)]$_{L2}$ + [DownSamp + ResBlock(256, 256, 512)]$_{L1}$ + [DownSamp + ResBlock(512, 512, 512)]$_{L0}$

Decoder parameters are as follows:

[MeshConv$^T$(512,512) + Concat + ResBlock(1024, 256, 256)]$_{L1}$ + [MeshConv$^T$(256,256) + Concat + ResBlock(512, 128, 128)]$_{L2}$ + [MeshConv$^T$(128,128) + Concat + ResBlock(256, 64, 64)]$_{L3}$ + [MeshConv$^T$(64,64) + Concat + ResBlock(128, 32, 32)]$_{L4}$ + [MeshConv$^T$(32,32) + Concat + ResBlock(64, 32, 32)]$_{L5}$ + [MeshConv(32,15)]$_{L5}$

Total number of parameters: 5180239

**Training details** Note that the number of output channels is 15, since the 2D3DS dataset has two additional classes (invalid and unknown) that are not evaluated for performance. We train our network with a batch size of 16, initial learning rate of $1 \times 10^{-2}$, step decay of 0.7 per 20 epochs, and use the Adam optimizer. We use the weighted cross-entropy loss for training. We weight the loss for each class using the following weighting scheme:

$$w_c = \frac{1}{1.02 + \log(f_c)} \tag{13}$$

where $w_c$ is the weight corresponding to class $c$, and $f_c$ is the frequency by which class $c$ appears in the training set. We use zero weight for the two dropped classes (invalid and unknown).

## B.4 CLIMATE PATTERN SEGMENTATION

**Architecture** We use the same network architecture as the Omnidirectional Image Segmentation task in Sec. B.3. Minor difference being that all feature layers are cut by 1/4.

Total number of parameters: 328339

**Training details** We train our network with a batch size of 256, initial learning rate of $1 \times 10^{-2}$, step decay of 0.4 per 20 epochs, and use the Adam optimizer. We train using weighted cross-entropy loss, using the same weighting scheme as in Eqn. 13.

## C DETAILED STATISTICS FOR 2D3DS SEGMENTATION

We provide detailed statistics for the 2D3DS semantic segmentation task. We evaluate our model's per-class performance against the benchmark models. All statistics are mean over 3-fold cross validation.

| Model | Mean | beam | board | bookcase | ceiling | chair | clutter | column | door | floor | sofa | table | wall | window |
|---|---|---|---|---|---|---|---|---|---|---|---|---|---|---|
| UNet | 0.5080 | 0.1777 | 0.4038 | **0.5914** | 0.9180 | 0.5088 | 0.4603 | 0.0875 | 0.4398 | 0.9480 | 0.2623 | 0.6865 | **0.7717** | 0.3481 |
| FCN8s | 0.4842 | 0.1439 | 0.4413 | 0.3952 | 0.8971 | 0.5244 | **0.5759** | 0.0564 | 0.5962 | **0.9661** | 0.0322 | 0.6614 | 0.7359 | 0.2682 |
| PointNet++ | 0.3349 | 0.1928 | 0.2942 | 0.3277 | 0.5448 | 0.4145 | 0.2246 | **0.3110** | 0.2701 | 0.4596 | **0.3391** | 0.4976 | 0.3358 | 0.1413 |
| Ours | **0.5465** | **0.1964** | **0.4856** | 0.4964 | **0.9356** | **0.6382** | 0.4309 | 0.2798 | **0.6321** | 0.9638 | 0.2103 | **0.6996** | 0.7457 | **0.3897** |

Table 6: Per-class accuracy comparison with baseline models

| Model | Mean | beam | board | bookcase | ceiling | chair | clutter | column | door | floor | sofa | table | wall | window |
|---|---|---|---|---|---|---|---|---|---|---|---|---|---|---|
| UNet | 0.3587 | 0.0853 | 0.2721 | 0.3072 | 0.7857 | 0.3531 | 0.2883 | 0.0487 | 0.3377 | **0.8911** | 0.0817 | 0.3851 | 0.5878 | **0.2392** |
| FCN8s | 0.3560 | 0.0572 | 0.3139 | 0.2894 | 0.7981 | 0.3623 | **0.2973** | 0.0353 | 0.4081 | 0.8884 | 0.0263 | 0.3809 | 0.5849 | 0.1859 |
| PointNet++ | 0.2312 | **0.0909** | 0.1503 | 0.2210 | 0.4775 | 0.2981 | 0.1610 | 0.0782 | 0.1866 | 0.4426 | **0.1844** | 0.3332 | 0.3061 | 0.0755 |
| Ours | **0.3829** | 0.0869 | **0.3268** | **0.3344** | 0.8216 | **0.4197** | 0.2562 | **0.1012** | **0.4159** | 0.8702 | 0.0763 | **0.4170** | **0.6167** | 0.2349 |

Table 7: Mean IoU comparison with baseline models

| # of parameters | 2558 | 6862 | 20828 | 70192 | 254648 | 960056 | 3737160 |
|---|---|---|---|---|---|---|---|
| Runtime (ms) | 2.9458 | 2.9170 | 2.8251 | 2.9222 | 3.0618 | 3.0072 | 2.9476 |

Table 8: Runtime analysis for classification network (as in ModelNet40 experiment in Sec. 4.2)

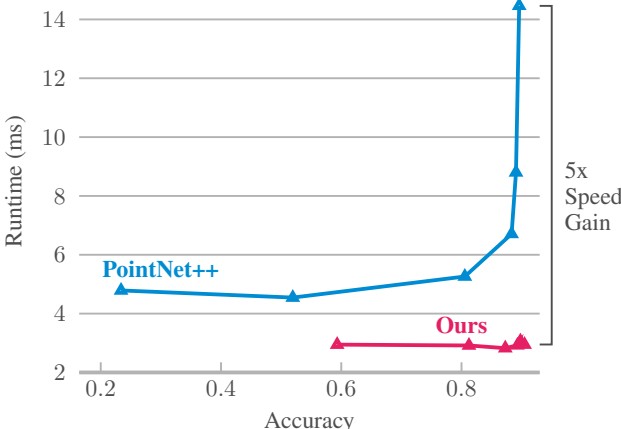

Figure 8: Comparison of runtime for our model and the only other model of comparable peak accuracy: PointNet++. Our model achieves a significant speed gain (5x) over the baseline in high accuracy regime.

## D    RUNTIME ANAYLSIS

To further evaluate our model's runtime gain, we record the runtime of our model in inference mode. We tabulate the runtime of our model (across a range of model sizes) in Table 8. We also compare our runtime with PointNet++, whose best performing model achieves a comparable accuracy. Inference is performed on a single NVIDIA GTX 1080 Ti GPU. We use a batch size of 8, and take the average runtime in 64 batches. Runtime for the first batch is disregarded due to extra initialization time. We observe that our model achieves fast and stable runtime, even with increased parameters, possibly limited by the serial computations due to network depth. We achieve a significant speedup compared to our baseline (PointNet++) of nearly 5x, particularly closer to the high-accuracy regime. A frame rate of over 339 fps is sufficient for real-time applications.

## E    IMPLEMENTION DETAILS FOR SEGMENTATION BASELINES

We provide further details for implementing the baseline models in the semantic segmentation task.

**FCN8S and U-Net**    The two planar models are slightly modified by changing the first convolution to take in 4 channels (RGBD) instead of 3. No additional changes are made to the model, and the models are trained from scratch. We use the available open source implementation[3] for the two models.

**PointNet++**    We use the official implementation of PointNet++[4], and utilize the same code for the examplar ScanNet task. The number of points we use is 8192, same as the number of points used for the ScanNet task. We perform data-augmentation by rotating around the z-axis and take sub-regions for training.

---

[3]https://github.com/zijundeng/pytorch-semantic-segmentation
[4]https://github.com/charlesq34/pointnet2

**S2CNN**     S2CNN was not initially designed and evaluated for semantic segmentation tasks. However, we provide a modified version of the S2CNN model for comparison. To produce scalar fields as outputs, we perform average pooling of the output signal only in the gamma dimension. Also since no transpose convolution operator is defined for S2CNN, we maintain its bandwidth of 64 throughout the network. The current implementations are not particularly memory efficient, hence we were only able to fit in low-resolution images of tiny batch sizes of 2 per GPU. Architecture overview:

[S2Conv(4, 64) +BN+ReLU]$_{b64}$ + [SO3Conv(64, 15)]$_{b64}$ + AvgPoolGamma

