# OpenReview forum: "Spherical CNNs on Unstructured Grids"
_ICLR.cc/2019/Conference_

### Official Review · AnonReviewer3 · 2018-10-21

**Rating:** 7
**Confidence:** 5

**Review:**

The paper presents a new convolution-like operation for parameterized manifolds, and demonstrates its effectiveness on learning problems involving spherical signals. The basic idea is to define the MeshConvolution as a linear combination (with learnable coefficients) of differential operators (identity, gradient, and Laplacian). These operators can be efficiently approximated using the 1-hop neighbourhood of a vertex in the mesh.

In general I think this is a strong paper, because it presents a simple and intuitive idea, and shows that it works well on a range of different problems. The paper is well written and mostly easy to follow. The appendix contains a wealth of detail on network architectures and training procedures.

What is not clear to me is how exactly the differential operators are computed, and how the MeshConvolution layer is implemented. The authors write that "differential operators can be efficiently computed using Finite Element basis, or derived by Discrete Exterior Calculus", but no references or further detail is provided. The explanation of the derivative computation is:
"The first derivative can be obtained by first computing the per-face gradients, and then using area-weighted average to obtain per-vertex gradients. The dot product between the per-vertex gradient value and the corresponding x and y vector fields are then computed to acquire grad_x F and grad_y F."
What are per-face gradients and how are they computed? Is the signal sampled on vertices or on faces? What area is used for weighting? What is the exact formula? What vector fields are you referring to? (I presume these are the coordinate vector fields). In eq. 5, what are F_i and F_j? What is the intuition behind the cotangent formula (eq. 5), and where can I read more? etc.

Please provide a lot more detail here, delegating parts to an appendix if necessary. Providing code would be very helpful as well.

A second (minor) concern I have is to do with the coordinate-dependence of the method. Because the MeshConvolution is defined in terms of (lat / lon) coordinates in a non-invariant manner, and the sphere does not admit a global chart, the method will have a singularity at the poles. This is confirmed by the fact that in the MNIST experiment, digits are rotated to the equator "to prevent coordinate singularity at the poles". I think that for many applications, this is not a serious problem, but it would still be nice to be transparent and mention this as a limitation of the method when comparing to related work.

In "Steerable CNNs", Cohen & Welling also used a linear combination of basis kernels, so this could be mentioned in the related work under "Reparameterized Convolutional Kernel".

To get a feel for the differential operators, it may be helpful to show the impulse response (at different positions on the sphere if it matters).

In experiment 4.1 as well as in the introduction, it is claimed that invariant/equivariant models cannot distinguish rotated versions of the same input, such as a 6 and a 9. Although indeed an invariant model cannot, equivariant layers do preserve the ability to discriminate transformed versions of the same input, by e.g. representing a 9 as an upside-down 6. So by replacing the final invariant pooling layer and instead using a fully connected one, it should be possible to deal with this issue in such a network. This should be mentioned in the text, and could be evaluated experimentally.

In my review I have listed several areas for improvement, but as mentioned, overall I think this is a solid paper.

---

> ### Author Response · Authors · 2018-11-26
> **Re: Review**
>
> Thank you for your thorough review and helpful comments. We will try to address your concerns and suggestions below:
> - Details on MeshConv
> We have added additional references as well as details for implementation of mesh differential operators in the Appendix. Additionally, we make our code anonymously available for reproducibility. Please check the code at the link below:
> https://drive.google.com/open?id=1z-hy3NVQtPxNcyDsRz-LqulwqDxNqAMo
>
> - Coordinate-dependence of the method (singularity at the poles).
> The method is coordinate dependent and coordinate singularity is an actual problem during implementation of the method. However, several tricks can be implemented to mitigate this issue. First, we use a spherical mesh subdivided from a base icosahedron that does not have a vertex that is at the pole. Then, all subsequent vertices will not be exactly residing on the pole, and numerically the singularity will not occur. Second, we always mute the signal at the poles (i.e., pad with zero). In practice this work extremely well, and tends not to affect the results. The major reason for rotating the spherical MNIST to the equator is in fact due to rotational invariance, since projecting the digits to the pole will turn the gradient components into radial and azimuthal ones, rendering the filters rotationally (around upward z axis) equivariant and the overall network invariant to rotations around z axis. We added discussions about its limitations in the revised paper.
>
> - Steerable CNNs
> Thank you for the suggestion. We have added the reference to the corresponding section.
>
> - Orientable models with equivariant layers
> Indeed equivariant convolutional operators do not prevent the network from being able to distinguish transformed versions of the same input. As per suggestion, we altered the original S2CNN network to be non-invariant by swapping the final global pooling layers with an average pool only in the gamma dimension (the extra dimension in SO(3) to for preserving equivariance), followed by a flattening operation in the spatial dimension. Furthermore, we added an additional fully-connected layer for enhanced representational power. Testing this network on MNIST dataset, we have the following findings:
> # of params: 162946
> Accuracy: 98.08
> Which has more parameters and lower accuracy than our proposed model. The experiment results suggest that since these equivariant operators are specifically engineered to preserve equivariance, they tend to not be the most efficient for orientable tasks that do not require equivariance.
> Additionally, to verify that orientability as been resolved, we compare the per-class accuracy for both the original S2CNN (rot-invariant version) and the modified S2CNN (not rot-invariant version). Below are the comparisons:
>
> Digit Class:               0       1       2       3       4       5       6       7       8       9
> ----------------------------------------------------------------------------------------------
> Original S2CNN:  0.99, 0.99, 0.98, 0.98, 0.96, 0.96, 0.98, 0.95, 0.96, 0.86
> Modified S2CNN: 0.99, 0.99, 0.98, 0.99, 0.97, 0.99, 0.99, 0.97, 0.97, 0.98
>
> Results show that removing the final pooling layer drastically improves accuracy for the digit “9” due to orientability, but overall lower accuracy compared to our spherical network suggests weaker representational power.
>
> - Visualizations
> We agree that visualizing the differential operators could be helpful for the reader. We have a visualization of an exemplary signal in Figure 1 that illustrates the differential operators.

---

### Official Review · AnonReviewer2 · 2018-11-02
**Simple and efficient model on spherical data, large scale experiments need more benchmarks**

**Rating:** 7
**Confidence:** 3

**Review:**

This article introduces a simple yet efficient method that enables deep learning on spherical data (or 3D mesh projected onto a spherical surface), with much less parameters than the popular approaches, and also a good alternative to the regular correlation based models.

Instead of running patches of spherical filters, the authors takes a weighted linear combination of differential operators applied on the data. The method is shown to be effective on Spherical MNIST, ModelNet, Stanford 2D-3D-S and a climate prediction dataset, reaching competitive/state-of-the-art numbers with much less parameters..

Less parameters is nice, but the argument could be strengthened if the authors could also show impressive results in terms of runtime. Typically number of parameters is not a huge issue for today’s deep networks, but for real-time robotics to be equipped with 3D perception, runtime is a much bigger factor.

I also think that the Stanford 2D-3D-S experiments have some issues:

UNet and FCN-8s are good baselines, but other prior work based on spherical convolution are omitted here. E.g. S2CNN and SphereNet. S2CNN has released their code so it should be benchmarked.

Additionally, comparison to PointNet++ could be a little unfair.

i) What is the number of points used in PointNet++? The author reported 1000 points for ModelNet which is ok for that dataset but definitely too small for indoor scenes. The original paper used 8192 points for ScanNet indoor scenes.

ii) Point-based can have data-augmentation by taking subregions of the panoramic scene, where as sphere-based method can only take a single panoramic image. The state-of-the-art method (PointSIFT) achieves ~70 mIOU on this dataset. PointNet(++) can also achieve 40-50 mIOU. Maybe the difference is at using regular image or panoramic images, but the panoramic image is just a combination of regular images so I wouldn’t expect such a large difference.

In conclusion, this paper proposes a novel deep learning algorithm to handle spherical data based on differential operators. It uses much less parameters and gets impressive results. However, the large scale experiments has some weaknesses. Therefore I recommend weak accept.

----
Small issues / questions:

- Notation lacks clarity. What are x, y in Eqn. 1? The formulation of convolution is not very clear to me, but maybe due to my lack of familiarity in this literature.

- In Figure 1, the terminology of “MeshConv” is first introduced, which should come earlier in the text to improve clarity.

- In the article, the author distinguished their method with S2CNN that their method is not rotation invariant. I don’t understand this part. In the architecture diagram, if average pool is applied across all spherical locations, then why is it not rotation invariant?

===
After rebuttal:
I thank the authors for addressing the comments in my review. It clarifies the questions I had about on the 2D3DS dataset (panorama vs. 3D points). Overall I feel this is a good model and have solid experiments. Therefore, I raise the score to 7.

---

> ### Author Response · Authors · 2018-11-26
> **Re: Simple and efficient model on spherical data, large scale experiments need more benchmarks**
>
> Thanks for your detailed and thorough review of our paper. We will try to address your questions and suggestions below:
> - Runtime
> We evaluate the runtime for our classification network and compare with the PointNet++ model which is of comparable peak performance. We report these runtimes in Appendix D. Our best performing model achieves a 5x speedup compared with PointNet++.
>
> - 2D3DS Baseline (add S2CNN)
> S2CNN and SphereNet were originally designed and evaluated for classification tasks. We corresponded with the authors of S2CNN and extended upon the original S2CNN architecture for semantic segmentation. We include the S2CNN results on 2D3DS dataset in the revised Figure 4. We detailed the modified S2CNN architecture in Appendix E. The best mIoU from the modified S2CNN model is significantly lower than ours (0.2581 vs 0.3829).
>
> - What is the number of points used in PointNet++?
> The number of points we use is 8192. We are using the same code from PointNet++ for the ScanNet task, where we do the data-augmentation by rotating around the z-axis and take subregions for the learning.
>
> - Difference between pano image and 3D point segmentation
> PointNet++ was initially designed for and tested on point clouds sampled from a 3D model that requires fusing multiple scans from various scan locations. Segmenting a single panorama, which is the setup in our experiment, is a much more challenging yet realistic task for engineering applications. A single view panorama poses multiple additional challenges, such as serious occlusions in the scene, noises in the depth map, and sparsity in the point cloud for objects that are far away from the viewpoint. We believe that all these problems can prevent the point-based method from achieving comparable results as in the original setup using uniformly sampled 3D points.
>
> - Notation lacks clarity
> Thank you for pointing out the issue in notation clarity. In this context, x and y refer to the spatial coordinates that correspond to the two spatial dimensions over which the convolution is performed. Eqn 1 through 3 states the fact that since convolution (and cross-correlation) operators are linear, traditional convolution operators can be viewed as linear combinations of the original signal convolved with the basis functions of the kernel.
>
> -  The terminology of “MeshConv”
> We have added the definition of this terminology to the introduction section, before its occurrence in Fig. 1.
>
> - Why is this method not rotationally invariant
> The method is not considered to be rotation invariant because the convolution operator is coordinate-dependent (depending on how the x-y coordinate vectors are defined on the manifold). Hence, the corresponding features will change due to a rotation, and the final pooled value will be different.

---

### Official Review · AnonReviewer4 · 2018-11-13
**Simple and effective idea.**

**Rating:** 6
**Confidence:** 3

**Review:**

Summary:
The paper proposes a novel convolutional kernel for CNN on the unstructured grids (mesh). Contrary to previous works, the proposed method formulates the convolution by a linear combination of differential operators, which is parameterized by kernel weights. Such kernel is then applied on the spherical mesh representation of features, which is appropriate to handle spherical data and makes the computation of differential operators efficient. The proposed method is evaluated on multiple recognition tasks on spherical data (e.g. 3d object classification and omnidirectional semantic segmentation) and demonstrates its advantages over existing methods.

Comments/suggestions:
I think the paper is generally well-written and clearly delivers its key idea/advantages. However, I hope the authors can elaborate the followings:

1) Analysis of computational cost
It would be helpful to elaborate more analysis on computational cost. The proposed formulation seems to involve the second-order derivatives in the backpropagation process (due to the first-order derivatives in Eq.(4)), which can be a computational bottleneck. It will be very useful to provide analysis on computational cost together with parameter efficiency study (Figure 3 and 4).

2) Intuitive justification
It would be great if the authors provide more intuitive descriptions on Eq.(4) (and possibly elaborate captions of Figure 1); what is the intuition of using differential operators? Why is it useful to deal with unstructured grids? How does it lead to improvement over the existing techniques?

Conclusion:
Overall, I think this paper has solid contributions; the proposed MeshConv operator is simple but effective to handle spherical data; the experiment results demonstrate its advantages over existing methods on broad applications, which are convincing. I think conveying more intuitions on the proposed formulation and providing additional performance analysis will help readers to understand paper better.

---

> ### Author Response · Authors · 2018-11-26
> **Re:Simple and effective idea.**
>
> Thank you for your response and constructive feedback! Below are some of our comments in response to your questions and suggestions:
> (1) Analysis of computational cost
> While the model computes second order spatial derivative (Laplacian) as a basis for the convolution operator, it ultimately amounts to a linear combination of these basis for the convolution step. Hence, training only involves first order gradients with respect to these weights to train (as opposed to using the Hessian). Generally, training time is difficult to benchmark as it involves many variables (hardware, DL framework etc.). However, we evaluate the inference runtime for our classification network and compare with the PointNet++ model which is of comparable peak performance. We report these runtimes in Appendix D. Our best performing model achieves a 5x speedup compared with PointNet++.
> (2) Intuitive justification
> We appreciate your feedback. We have added more intuitive explanations to our paper in Sec 1 as well as the captions of Fig. 1.

---

### Public Comment · (anonymous) · 2018-10-07
**Simplistic criticism against equivariant architectures**

In the introduction you say "[...] assumed orientation information is crucial to the predictive capability of the network [...] omnidirectional images, where images are naturally oriented by gravity [...]".
Let me inform you that there is a simple trick to solve this problem: add an extra input feature map that indicates the orientation of the gravitational field.

Indeed if the symmetry completely broken like for the example of MNIST then you better have to give up the equivariant architecture. But for tasks when the symmetry is only partially broken like planets oriented by their axis of rotation then equivariant architectures are still relevant and the axis of rotation can be given as part of the input.

---

> ### Author Response · Authors · 2018-10-07
> **Orientability**
>
> Thank you for  your feedback and your interest in our paper! We would like to clarify our wording of this statement. Admittedly various current equivariant architectures can be made into non-equivariant counterparts with additional enhancements such as additional feature layers. However such enhancements would render the equivariant architectures  into non-equivariant ones, therefore our general statement that "assumed orientation information is crucial to the predictive capability of the network (for a range of problems)" is nevertheless accurate.
>
> Also, as a side note only for further discussion, equivariant architectures have a particular construct to maintain equivariance (such as adding an additional dimension for SO(3) layers in S2CNN), and tend not to be most efficient for orientable tasks.

---

### Meta-Review · Area_Chair1 · 2018-12-14
**decision**

**Confidence:** 4
**Recommendation:** Accept (Poster)

**Metareview:**

The paper presents a simple and effective convolution kernel for CNNs on spherical data (convolution by a linear combination of differential operators). The proposed method is efficient in the number of parameters and achieves strong classification and segmentation performance in several benchmarks. The paper is generally well written but the authors should clarify the details and address reviewer comments (for example, clarity/notations of equations) in the revision.